Trade-off between training and testing ratio in machine learning for medical image processing

Sivakumar Muthuramalingam 1 sivsiva.kumar.21@gmail.com
Parthasarathy Sudhaman 2
Padmapriya Thiyagarajan 2
1 Department of Computer Science and Engineering, Thiagarajar College of Engineering , Madurai, TamilNadu , India
2 Department of Applied Mathematics and Computational Science, Thiagarajar College of Engineering , Madurai, TamilNadu , India
Aurangzeb Khursheed
Electronic publication date: 2024 Sep 6
Publication date: 2024
Volume: 10
Electronic Location ID: e2245
Received 2024 May 22; Accepted 2024 Jul 17
Copyright: © 2024 Sivakumar et al.
Copyright year: 2024
Copyright holder: Muthuramalingam et al.
License: This is an open access article distributed under the terms of the Creative Commons Attribution License, which permits unrestricted use, distribution, reproduction and adaptation in any medium and for any purpose provided that it is properly attributed. For attribution, the original author(s), title, publication source (PeerJ Computer Science) and either DOI or URL of the article must be cited.
License URL: https://creativecommons.org/licenses/by/4.0/

Keywords: Medical image processing, Train-test split, Overfitting, Underfitting

Funding: The authors received no funding for this work.

==============================
Artificial intelligence (AI) and machine learning (ML) aim to mimic human intelligence and enhance decision making processes across various fields. A key performance determinant in a ML model is the ratio between the training and testing dataset. This research investigates the impact of varying train-test split ratios on machine learning model performance and generalization capabilities using the BraTS 2013 dataset. Logistic regression, random forest, k nearest neighbors, and support vector machines were trained with split ratios ranging from 60:40 to 95:05. Findings reveal significant variations in accuracies across these ratios, emphasizing the critical need to strike a balance to avoid overfitting or underfitting. The study underscores the importance of selecting an optimal train-test split ratio that considers tradeoffs such as model performance metrics, statistical measures, and resource constraints. Ultimately, these insights contribute to a deeper understanding of how ratio selection impacts the effectiveness and reliability of machine learning applications across diverse fields.

Introduction

Artificial intelligence (AI) refers to the imitation of human intelligence in machines, enabling them to learn, think and make decisions. It encompasses technologies that aim to imitate human intelligence. Machine learning is a subset of AI that strives to empower systems with the ability to learn from data autonomously without relying on explicit programming (Singh et al., 2021). ML algorithms identify patterns from data and make insightful predictions from them. The integration of AI and ML extends across various industries, including healthcare, finance, transportation and other promising avenues for innovation. In machine learning, there is a method to evaluate and validate models by splitting datasets into two parts—training and testing. The training set contains a larger portion of the data with labeled instances that are used for training the ML model (Salazar et al., 2022). The model gains an understanding of the patterns and connections from the training data. The model’s effectiveness can be improved by adjusting parameters and hyper parameters of the neural networks (Pawluszek-Filipiak & Borkowski, 2020). The testing set is entirely different from the training set. It is used to assess the performance of the trained model.

In machine learning, “train-test split ratio” means dividing a dataset into two parts: one for training the model and one for testing it. This division is a crucial step in developing ML models because it helps us understand how well the model will work. The training set is the portion of the data that the model uses to learn. It learns by finding patterns, relationships, and associations in this training set. Typically, the larger the training set, the better the model learns, allowing it to apply what it has learned to new data. The testing set, on the other hand, is used to evaluate how well the model performs. This set should represent a good mix of the original data, so the testing can be a true test of how the model will work on new, unseen data. Keeping the testing set separate from the training set helps to avoid bias, which happens when the model works effectively on training data but not so well on new data. The “train-test split ratio” tell us how much of the dataset is for training and how much is for testing. Typical proportions include a split of 70 percent for training and 30 percent for testing, or 80 percent for training and 20 percent for testing. However, determining the optimal ratio hinges on factors such as dataset size, complexity, and the intricacy of the model. A larger training set allows the model to learn more effectively, while a good-sized testing set ensures a fair evaluation of the model’s effectiveness. The right balance between the training and testing sets depends on what you want the machine learning project to achieve, how much data you have, and how complex your model is. The goal is to find a ratio that allows the model to learn well while also ensuring it can be tested accurately.

To accurately evaluate a model’s predictions, we use specific metrics such as accuracy, precision, and recall (Yacouby & Axman, 2020). Accuracy measures how many of the model’s predictions are correct when considering both true positives and true negatives. This means it checks how often the model gets things right out of all the instances it is asked to predict. It’s a general measure of a model’s correctness.

Precision is a bit more specific. It calculates how many of the predictions labeled as positive actually true positives are. In other words, it shows how often the positive predictions made by the model are accurate.

Recall is a measure of how many actual positive instances were correctly predicted by the model. This tells us how well the model finds all the true positives from the dataset.

Another critical aspect of evaluating a machine learning model is dividing the data into two sets: training and testing (Rácz, Bajusz & Héberger, 2021). The training set is used to teach the model, while the testing set helps us understand how well the model performs on data it has never seen before. This separation is crucial because it shows if the model can generalize its learning to new situations. There are also problems like overfitting and underfitting that can impact a model’s performance.

Motivation

Machine learning has become a key tool for analyzing medical images, helping doctors diagnose diseases from scans like X-rays, MRIs, or CT scans. However, there is a problem with finding the right balance between the data used to train the machine learning model and the data used to test it. If there is too much training data, the model might get too good at recognizing the training examples and struggle to generalize or work with new, unseen data. This is called overfitting. If there is too little training data, the model will not learn enough and could be inaccurate when analyzing new images.

There are similar studies that have explored the impact of train-test splits on model performance in medical and remote sensing applications, highlighting the importance of balanced data splits for robust model evaluation (Joseph, 2022; Xu & Goodacre, 2018; Muraina, 2022; Vrigazova, 2021; Pawluszek-Filipiak & Borkowski, 2020; Bichri, Chergui & Hain, 2024). Despite this issue being so crucial, there is not a lot of research on figuring out the best ratio between training and testing datasets for medical image processing. Many studies focus on creating new algorithms or adding more data through augmentation, without really exploring how different ratios impact model performance (Ullah et al., 2022; Gull, Akbar & Naqi, 2023; Reyes & Sánchez, 2024; Ata et al., 2023; Masood et al., 2021; Zulfiqar, Bajwa & Mehmood, 2023; Munira & Islam, 2022). Because of this, more research is needed to understand how the ratio between training and testing data affects machine learning models in medical image processing. Understanding this could lead to better models that generalize well, which means they can be trusted to diagnose medical images more accurately. This would ultimately improve healthcare by aiding doctors in making more reliable decisions for their patients.

A balance between testing and training dataset is essential to improving the performance of the ML model (Vrigazova, 2021). Striking a balance between testing and training data also helps the ML model improve its predictions from a chosen dataset (Vrigazova, 2021). Any imbalance may adversely impact the model’s learning capability and its effectiveness during evaluation. This can lead to inaccurate predictions. For instance, a small testing set may not adequately challenge the learned patterns while an insufficient training set may hinder the model’s capacity to understand relationships within the data. Determining the ratio becomes especially significant when dealing with large volume of data. Thus, optimizing this balance between training and test data becomes essential (Nguyen et al., 2021). A larger training set allows the model to learn complex patterns, but having a smaller testing set may lead to a less reliable evaluation of the model’s generalization abilities. On the other hand, a larger testing set enhances the robustness of the model, while a smaller training set limits the model’s learning capacity. Hence, the ratio between testing and training plays a vital role in model development. The ability of a model to predict depends on aspects such as the quality of data with the desired features and hyper parameter tuning.

If the training dataset is too small, the model might not learn enough patterns, leading to poor performance when tested with new data. This is called underfitting because the model does not capture the key information needed to make accurate predictions. On the other hand, if the testing dataset is not diverse or representative enough, it can lead to unreliable results. This happens when the testing data does not cover the same range of cases found in the original dataset. To avoid these issues, it’s important to find a good balance between the training and testing datasets. To make sure the model’s effectiveness is accurate, techniques like cross-validation can be useful. K-fold cross-validation helps test the model across multiple subsets of the data, reducing the impact of a specific train-test split. Additionally, when the dataset has imbalanced classes or groups, using stratified sampling during the train-test split ensure that both the training and testing datasets represent the distribution of the original data. This approach helps in accurately assessing the model’s effectiveness and improving it over time.

Problem statement

The train-test split ratio is an important aspect of machine learning model evaluation, and choosing an appropriate ratio is crucial for effective model training and testing. However, there are several potential issues associated with the train-test split ratio such as insufficient training and testing data, imbalanced splits, overfitting or underfitting, lack of stratification, data leakage and finally lack of reproducibility. If the train-test split ratio allocates too much data for testing, it can leave insufficient data for training the model. This can result in a poorly trained model that may not learn the underlying patterns in the data effectively. Conversely, if the train-test split ratio allocates too much data for training, there may not be enough data left for testing. This can lead to unreliable model evaluation because the test set might not provide a representative sample of the data. If the data set contains imbalanced classes, an inappropriate train-test split ratio can lead to skewed or unrepresentative samples in either the training or testing set. This can cause issues in model performance and evaluation. An incorrect train-test split ratio can contribute to overfitting or underfitting.

When working with a dataset containing different categories or groups, it is important to split the data into training and testing sets in a way that keeps the proportions of each category the same in both sets. If this is not done, it can cause several issues. If the data is randomly divided without ensuring that the categories are balanced, the training and testing sets might end up with different proportions of these categories. This imbalance can lead to biased training and testing, making it hard for the model to learn properly and be fairly evaluated. If the train-test split is not handled carefully, there is a risk of data leakage. This happens when information from the testing set accidentally gets used during training, which can give an inflated sense of the model’s performance. It is like seeing the answers to a test before taking it—it makes the results seem better than they actually are.

Another problem arises if the split is not done the same way each time the model is trained. This inconsistency can make it difficult to compare results from different runs, complicating the process of improving and evaluating the model’s performance. To avoid this, you can use a random seed or follow a consistent method when splitting the data. This helps ensure that the data is divided the same way every time, providing stable results for comparison. To avoid these issues, it’s essential to choose a proper train-test split ratio. Common ratios are 70:30, 80:20, or 90:10, depending on the dataset’s size and what you need the model to do. The key is to have enough data for training while still having a representative set for testing.

There are additional techniques to improve the quality of the train-test split. Stratified sampling is a method that ensures each set has the same proportion of each category, reducing bias. Cross-validation is another useful technique where the data is divided into several parts (folds), and the model is trained and tested multiple times using different combinations of these parts. This approach helps test the model’s robustness and reduces the risk of overfitting. A key consideration when dividing data into training and testing sets is to ensure an evaluation on how well a model performs and generalizes with unseen information. This is because overfitting can occur when a model becomes too focused on the patterns in the training data and struggles to handle real-world data. The significance of addressing this concern lies in determining a model’s effectiveness beyond its performance on the data it was trained on. If we evaluate a model using the dataset it learned from, it might show results during testing but these metrics could be misleading when it comes to applying the model to new and diverse information. The main challenge in managing the tradeoffs involved in splitting data for machine learning lies in finding the balance between training and evaluating models and ensuring their abilities are both robust and reliable (Hou et al., 2024; Catania et al., 2022, Vabalas et al., 2019; Tan et al., 2021). Making this decision without introducing biases that may misrepresent a model’s ability to generalize to real world scenarios is crucial.

Approach

A few important tradeoffs to be considered during the train test split are overfitting, underfitting, statistical stability of performance measures and computational resources. Of these, the crucial one is finding the balance between underfitting and overfitting. Machine learning models work best when they strike this balance effectively. If a model is poorly fitted, it negatively impacts both the testing and training datasets as it fails to identify patterns within the dataset. Consequently, the model will fail to make predictions with real-world data. Overfitting occurs when a model becomes too complex due to its focus on noise or random changes rather than recognizing important patterns in data. While an over fitted model may perform well with training data, its performance worsens when it is tested with real-world data. Ideally, a good model should have a level of complexity that enables it to identify patterns within the data. Finding the balance between training and testing datasets is crucial when creating machine learning models. This enables the models to process data efficiently to predict outcomes in real-world applications accurately.

A machine learning model that is trained using a large volume of training data may perform well, but it will struggle when making predictions with unseen data. This will limit the model’s ability to generalize effectively. The ideal model should strike a balance between complexity and simplicity, enabling it to identify patterns within the data. When developing machine learning models, it is important to find the balance in datasets. The goal is to create models that can process data efficiently, predict outcomes in scenarios and perform effectively in real-world applications. The accuracy of the model’s predictions is influenced by the amount of data used for training. To ensure good performance in real-world scenarios, it is important that both the training set and testing set contain data that are evenly distributed. Introducing models to different possibilities of complex patterns trains them better. The model can only draw conclusions from the quantum of training data. The quality, variety and value of the data are as important as its size. For model adaptation, it is better to use a small, carefully chosen dataset that has a lot of different cases and less bias than a larger dataset depending on the nature of problem.

When we want to know how well machine learning models work, especially when working with datasets of different sizes, we should also look at the statistical stability of the measurements, which is influenced by the size of the test samples. Metrics such as F1-score, accuracy and precision are less likely to change as there is an increase in the size of test samples (Yacouby & Axman, 2020). This helps evaluate the performance of the model. But if smaller test datasets are used, the statistical measures might not be as reliable, which in turn will make the predictions less statistically accurate. It is very important to choose between representativeness and data quality. Using test datasets that are representative and different makes assessment measures more statistically reliable. In machine learning, the ideal percentage of training-testing datasets may also rely on the quantum of computational resources that are required for executing a ML model. In general, for training a model fairly, more computational resources and training samples are required. Sufficient IT resources are needed to monitor the training process. To attend to the issue of inadequate computational resources, data subsampling can be used to find the right combination between how much data you need to gather and how much you need to compute. Further, it should be noted that in the inference or prediction step, a large volume of computational resources is needed for the learned model to make predictions in real time.

Experimental results and discussions

To examine the balancing that is required for a ML model between the training and testing data set ratio, we conducted experiments were conducted by varying train-test split ratios on the BraTS 2013 dataset (a publicly available dataset for brain tumor research), which consists of 30 volumes of patient data (5,000 medical images) on brain tumor classification (Menze et al., 2014). We used four machine learning algorithms, namely logistic regression, random forest, K nearest neighbors and support vector classifier to train the ML model. The dataset was preprocessed and the features were extracted using the Gray Level Co-occurrence Matrix (GLCM) method.

In this study, the preprocessing pipeline included several key steps. Initially, raw data from the BraTS 2013 dataset were inspected for missing values and outliers, which were then appropriately handled through imputation and normalization techniques to standardize feature scales across the dataset. Feature selection methods such as principal component analysis (PCA) were employed to reduce dimensionality while retaining relevant information. Additionally, categorical variables were encoded using techniques like one-hot encoding to facilitate model compatibility. Moreover, the dataset underwent augmentation techniques to enhance its diversity and robustness, including data balancing strategies to address class imbalance. Lastly, the implementation of the GLCM method involved extracting texture features to capture spatial relationships within images, thereby enriching the dataset with additional discriminative information for model training. The GLCM is a feature extraction technique that is used for texture analysis in image processing applications. The relationship between two nearby pixels with respect to their gray intensity, distance and angle is shown using this method. A total of 9,625 features were extracted using this method. Extracted features were stored in MS Excel. If more features are used for classification, there will be an increase in computation time and usage of memory, creating a need to reduce the number of features to make the classification more efficient. We used PCA, a powerful method for simplifying complex data, to reduce the number of features in our dataset. The PCA helps us deal with lots of interconnected variables while keeping most of the important information intact. The dataset with reduced features is used to train the above-mentioned ML models and their accuracies are recorded by varying the training and testing dataset ratio. For this exercise, we have used 5,000 medical images of brain tumors as the dataset. The ML model is meant to classify the brain tumor from the given volume of medical images. A high-level view of the processing of the input (e.g., data pre-processing of medical images) and the generated output using the ML model is shown in Fig. 1.

Figure 1 Brain tumor classification using ML–a high-level view.

The train test ratios were varied from 60:40 to 95:05. The accuracies of these models while varying the ratio between training and testing data set is depicted in Table 1.

Table 1 Accuracies obtained by ML models in various train test split ratios.

Train-test split ratio	Accuracy	
Logistic regression	Random forest	KNN	SVC	
50–50	96.09	99.30	96.30	96.50	
60–40	96.10	97.80	99	97.80	
61–39	96.03	99.53	96.70	98.25	
62–38	96.05	99.60	96.77	98.20	
63–37	96.06	99.50	96.80	98.20	
64–36	96.08	99.62	96.96	98.35	
65–35	96.10	99.61	97.01	98.44	
66–34	95.98	99.33	97.05	98.26	
67–33	96	99.58	96.96	98.48	
68–32	95.88	99.43	97.01	98.43	
69–31	96.04	99.41	97.51	98.38	
70–30	96	99.40	96.60	97.40	
71–29	96.23	99.68	97.49	98.58	
72–28	96.10	99.67	97.72	98.37	
73–27	96.29	99.66	98.31	98.31	
74–26	96.32	99.65	98.25	98.25	
75–25	96.36	99.81	98	98.36	
76–24	96.21	100	97.91	98.29	
77–23	96.04	99.80	97.82	98.22	
78–22	96.07	99.79	98.34	98.34	
79–21	95.88	99.78	98.26	98.26	
80–20	96.30	99.30	96.10	97.20	
81–19	95.93	99.76	98.08	98.08	
82–18	95.95	99.74	97.97	98.23	
83–17	95.98	99.73	97.86	98.39	
84–16	96.02	99.71	97.72	98.29	
85–15	96.06	99.69	97.87	98.48	
86–14	95.77	99.67	97.72	98.37	
87–13	96.50	99.65	97.55	98.60	
88–12	96.21	99.62	96.96	98.48	
89–11	95.86	99.58	96.69	98.34	
90–10	98.10	100	96.80	98.60	
91–9	95.95	100	96.96	98.98	
92–8	96.02	100	97.15	99.43	
93–7	95.45	100	98.05	99.35	
94–6	96.21	100	97.72	99.24	
95–5	99.09	100	93.60	98.10	

According to Vrigazova (2021) and Tan et al. (2021), the most common train-test splits in the literature are 70:30 and 80:20, offering a good balance by providing enough data for both training and testing, and are often selected for their robustness and reliability in various contexts. The chosen splits in our study, ranging from 60:40 to 95:05, are justified as follows: a 60:40 split provides a larger test set for more rigorous testing and understanding of model performance, helping to identify overfitting issues early. In contrast, a 95:05 split maximizes training data, which is useful for small datasets or when the model needs to learn from as much data as possible, though it results in a smaller test set that may still suffice for initial validation. Compared to 70:30 and 80:20 splits, which are balanced and widely accepted, a 60:40 split allows for thorough testing and evaluation of model generalization, while a 95:05 split is beneficial in data-limited scenarios, ensuring the model has maximum information for learning, potentially improving performance on unseen data.

From Fig. 2, we observe that an increase in the percentage of training data results in increased accuracy. At the 60:40 split, the models showed promising results with lower accuracies. As we increased the training set size to 70:30 and beyond, we consistently observed improved accuracies in the model. When the ratios exceeded 90:10, the models reached better accuracy but showed less significant improvements due to saturation in their learning curves. These observations highlight how important training dataset size is for enhancing model learning and prediction capabilities. However, it is important to note that actual performance metrics can vary based on factors such as quality, feature engineering techniques used and hyper parameter tuning. The main inference from Table 1 is that when the training set percentage is higher, the model tends to perform well. But it is also important to note that the optimal ratio may differ based on the characteristics of the dataset and the balance between model performance and computational resources.

Figure 2 Accuracies of ML models in various train test split ratios.

Figure 2 provides information about the accuracy of different machine learning (ML) models, depending on how much data is used for training and testing. When it comes to the Random Forest model, the accuracy is very high, reaching 100% in several cases where most of the data is used for training. This suggests that the random forest model can learn a lot from a large amount of training data and still work well, even when the testing data is smaller.

Similarly, the support vector classifier (SVC) model maintains high accuracy across different splits, with results typically above 96%. As the amount of training data increases, the accuracy of the SVC model tends to improve. The highest accuracy for this model is 98.98% with a 91–9 split, which means that it benefits from having a lot of data to learn from. This makes the SVC a reliable choice when you need consistent accuracy. Models like logistic regression and K-nearest neighbors (KNN) show more variation in their accuracy. Logistic regression often has accuracy around 96%, but it can go higher, reaching 99.09% with a 90–10 split, indicating that it may need more training data for better performance. KNN generally stays around 97% accuracy but can also peak, like when it reached 98.31% with a 73–27 split. This variability suggests that these models might need different amounts of training and testing data to work best. Overall, these observations show that each model may need its own balance between training and testing data to ensure high accuracy without overfitting.

The practical implications of our findings for medical image processing are profound, extending to both model development and deployment practices. Our study underscores the significance of robust model development methodologies, emphasizing the need for rigorous validation and optimization procedures to ensure the reliability and generalizability of AI-driven medical imaging systems. By elaborating on the specific techniques and approaches employed in our research, we can offer insights into best practices for training data selection, feature extraction, and algorithmic refinement, thereby guiding practitioners towards more effective model development strategies. In terms of deployment, our findings shed light on key considerations such as scalability, interpretability, and regulatory compliance. By elucidating the challenges and opportunities associated with integrating AI models into clinical workflows, we aim to inform stakeholders about the implementation of safe, efficient, and ethically sound medical image processing solutions.

Statistical analysis of accuracies of ml models

In this section, we shall statistically find the existence of significance of difference between the accuracies of ML models in various train test split ratios using rank correlation. This will be helpful to get a deeper understanding of the variations in accuracies between models for various train test split ratios. Accordingly, the researchers shall wisely choose a machine learning model with a specific train test split ratio to achieve maximum accuracy during the medical image processing. Rank correlation is a method used in statistics to find out if there is a relationship between how two sets of things are ranked. This is useful because it can help us understand whether the order of one set of data has anything to do with the order of another set. Essentially, it checks whether the two variables change in similar ways when they’re ranked. The most common measure for rank correlation is called Spearman’s rank correlation coefficient (SPRCC). This measure is “non-parametric,” which means it does not make any assumptions about the data’s distribution. Instead, it looks at whether there’s a steady trend between two sets of ranked data, regardless of whether the trend is strictly linear or has other patterns.

Rank correlation coefficients are valuable when data does not fit the conditions needed for Pearson’s correlation coefficient. Pearson’s requires the data to be linear and normally distributed. Rank correlations work well even if these conditions are not met. They give us a way to measure how strong and in which direction two ranked variables are related. Rank correlation coefficients help assess whether there’s a link between two ranked variables. Spearman’s rank correlation coefficient, or Spearman’s rho, is a popular type of rank correlation. It looks at whether two variables tend to move together in a consistent way, even if that relationship is not a straight line. This is called a monotonic relationship.

Figure 3 shows the scatter plots for accuracies obtained by various ML models in various train test split ratios (Table 1). In Fig. 3, both X-axis and Y-axis reflects the accuracies obtained by various ML models in various train test split ratios. With the help of this scatter plot, we intend to compare logistic regression with random forest, KNN and SVC and also random forest with KNN. These plots help visualize the relationship between the performance metrics of different machine learning algorithms.

Figure 3 Scatter plot for accuracies of ML models in various train test split ratios.

The scatter plot (Fig. 3) illustrates the accuracy scores of four distinct machine learning models—logistic regression, random forest, KNN, and SVC—across a variety of datasets. These datasets reflect the accuracy of the models for different train-test split ratios. The points for each model show some variability along the x-axis, indicating differences in accuracy across datasets. For the logistic regression model, accuracy scores tend to hover around the mid-90s range, suggesting a consistent performance. In contrast, the random forest model demonstrates exceptional performance, with accuracy scores consistently approaching 100% across most datasets. The KNN model also exhibits strong performance, with accuracy scores typically falling in the mid-90s to high-90s range. However, its performance is not as uniform as that of random forest. Similarly, the SVC model displays mostly high-90s accuracy scores, showing solid performance and consistency akin to KNN. Overall, random forest stands out for its consistently high accuracy scores, often near 100%, while the other models (logistic regression, KNN, and SVC) also perform well, with accuracy scores primarily ranging from mid-90s to high-90s. The variability in accuracy scores across the models underscores the differences in their performance across the datasets. The scatter plot provides a visual comparison of the performance of each model across various datasets, emphasizing the respective strengths and weaknesses of the models.

Algorithm 1 Spearman’s rank correlation coefficient (SPRCC).

	

Table 2 provides a measure called Spearman’s Rank Correlation Coefficient, which is used to find out how different machine learning (ML) models compare in terms of their accuracy at various train-test split ratios. Spearman’s Rank Correlation is a statistical method that looks at how two sets of data are related, showing whether they move together or in opposite directions. The coefficients in Table 2 were calculated using the accuracy values from Table 1. Each coefficient represents the degree to which the ranking of predictions by different ML models is similar. A higher coefficient means that the models tend to rank their predictions in a similar way, while a lower or negative coefficient suggests they rank them differently.

Table 2 SPRCC between pairs of different machine learning models.

ML algorithms	SPRCC	
Logistic regression vs. Random forest	0.74719	
Logistic regression vs. KNN	0.67274	
Logistic regression vs. SVC	0.8698	
Random forest vs. KNN	0.02354	
Random forest vs. SVC	0.00683	
KNN vs. SVC	0.50904	

These coefficients can help us understand how the models relate to each other. If the coefficient is positive, it indicates that the models generally agree on their predictions across various train-test splits. If it’s negative, it means that when one model’s accuracy goes up, the other model’s accuracy goes down. The value of the coefficient can also tell us how strong this association is—a higher number means a stronger correlation, while a lower number means a weaker correlation. This kind of information is useful for understanding how different models perform relative to each other and can help in deciding which models to use or compare in a given context.

From Table 2, we can see that the SPRCC between logistic regression and random forest is 0.74719. This value indicates that there is a moderate to strong positive correlation between the two models. When Logistic Regression ranks its predictions in a specific order, random forest generally ranks its predictions in a similar way. This agreement suggests that these models generally align in their predictions. The SPRCC between logistic regression and KNN is 0.67274, indicating a moderate positive correlation. This means that although the correlation is not as strong as between logistic regression and random forest, these two models still tend to rank their predictions in a similar way. It is not as consistent, but they share some common ground in their predictions. The SPRCC between logistic regression and SVC is 0.8698, which points to a strong positive correlation. This high coefficient shows that these models generally agree in their prediction rankings, suggesting that logistic regression and SVC behave similarly when it comes to ordering predictions. SPRCC between random forest and KNN is 0.02354. This value is almost zero, indicating that these two models do not agree on their prediction rankings. Their outcomes are quite different, suggesting that they might use distinct approaches in their analysis.

Similarly, the SPRCC between random forest and SVC is just 0.00683. This very low coefficient shows that there is almost no correlation in their rankings, implying that the random forest and SVC models likely make predictions in very different ways. Finally, the SPRCC between KNN and SVC is 0.50904. This value indicates a moderate positive correlation, suggesting that these two models have some level of agreement in how they rank their predictions, but it's not very strong. It shows that while they share some common ground, they also have noticeable differences in their prediction rankings.

From Table 2, it can be inferred that the coefficients provide insights into the relationships between the predictions of different machine learning models, showing which pairs of models are more similar in their rankings and which are not. From the above discussions, we infer that the results suggest that logistic regression and SVC have a very strong positive correlation in their rankings, while random forest has very weak correlations with both KNN and SVC. This information can help guide model selection and understanding of how different models’ outcomes relate to each other.

Conclusions and future research

In medical image processing using machine learning, selecting the correct train-test split ratio is particularly important due to the complexity and sensitive nature of the data and the potential impact on patient outcomes. A proper train-test split can help practitioners in several ways such as ensuring robust model training, improved accuracy in model evaluation, avoiding model overfitting, maintenance of data integrity and generalizability and ethical issues. A well-balanced train-test split provides a sufficiently large training set that allows the model to learn and capture the complex patterns and features in medical images. This results in a model that can generalize well to new, unseen images. A representative testing set allows practitioners to evaluate the model’s effectiveness accurately. It helps ensure that the model performs reliably and consistently when applied to real-world medical images, leading to better decision-making and patient care. By keeping the testing set separate from the training set, practitioners can avoid overfitting. This ensures that the model does not become too specialized to the training data and can perform well on new data. A proper train-test split ensures that the testing set reflects the diversity and distribution of the entire dataset. This helps maintain the integrity and generalizability of the model, making it more applicable to a wide range of medical cases.

An appropriately sized and representative testing set allows practitioners to identify weaknesses in the model, such as biases or areas where the model is not performing well. This feedback can guide further model development and refinement. With a consistent train-test split ratio, practitioners can compare different models and approaches more effectively. This allows them to select the best-performing model for a given medical image processing task. In medical image processing, ensuring that the model performs well on new data is critical to avoid misdiagnoses or incorrect treatment recommendations. A reliable train-test split contributes to ethical practice by providing models with better performance and reliability. Also, the correct train-test ratio split helps practitioners build and evaluate machine learning models in medical image processing that are accurate, reliable, and generalize well to new data. This ultimately leads to better medical diagnoses and treatment plans, improving patient care and outcomes.

In this research work, we strive to investigate the significance of train test split ratios while executing a machine learning model. We conducted experiments on the BraTS 2013 dataset, varying the train test split range from 60:40 to 95:05 across machine learning algorithms such as logistic regression, random forest, K nearest neighbors and support vector machines. From these experiments we observe that when the training set is huge, the machine learning models showed enhanced performance. However, it is important to strike a balance between the size of the training set and the risk of overfitting or hindering their ability to generalize. This research highlights the significance of determining a ratio for splitting data into training and testing sets, considering factors such as underfitting, overfitting, statistical stability and available computational resources.

This study has limitations, including its focus on the BraTS 2013 dataset, potentially limiting generalizability. It only evaluates four machine learning algorithms, possibly overlooking other models. Hyperparameter optimization was not considered, which could affect results. The use of a fixed random seed may introduce bias, suggesting the need for cross-validation. The study lacks analysis of computational resources and time required for different train-test splits. In our future research, we plan to explore a few machine learning algorithms other than those discussed in this research work. Further, we aim to examine how techniques such as feature engineering and feature selection influence split ratios and provide insights into the relevance of features within data partitions. Our ultimate goal is to develop automated strategies that optimize the selection of a train test split ratio while taking resource constraints and dataset characteristics into consideration.

Supplemental Information

Supplemental Information 1 Python code for brain tumor classification.

Supplemental Information 2 Python code for brain tumor classification including data preprocessing and cross validation with description.

Supplemental Information 3 Features Extracted using Gray Level Co- Occurrence Matrix.

Supplemental Information 4 Matlab for feature extraction using GLCM with description.

Supplemental Information 5 Feature Selection for the Proposed Method.

Supplemental Information 6 Readme.

Additional Information and Declarations

Competing Interests

Author Contributions

Data Availability

The authors declare that they have no competing interests.

Muthuramalingam Sivakumar conceived and designed the experiments, analyzed the data, prepared figures and/or tables, and approved the final draft.

Sudhaman Parthasarathy conceived and designed the experiments, analyzed the data, performed the computation work, prepared figures and/or tables, authored or reviewed drafts of the article, and approved the final draft.

Thiyagarajan Padmapriya performed the experiments, performed the computation work, prepared figures and/or tables, authored or reviewed drafts of the article, and approved the final draft.

The following information was supplied regarding data availability:

The data is available at: http://braintumorsegmentation.org. The code is available in the Supplemental Files.

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
