# Peer review of "Trade-off between training and testing ratio in machine learning for medical image processing"

_PeerJ Computer Science, doi:10.7717/peerj-cs.2245_

## Round 0.1 · original submission · Minor Revisions

The review comments of the expert reviewers are provided to improve its quality further.

Reviewer 1 ·

Basic reporting

Manuscript Review
Pros:
• The manuscript is clearly written and structured, making it accessible to a broad audience.
• It provides a thorough background on the importance of train-test split ratios and their impact on model performance, especially in medical image processing.
• The use of multiple machine learning models enhances the comprehensiveness of the study.
Cons:
• More citations to recent studies would strengthen the literature review, providing better context and validation for the research. For instance, similar studies have explored the impact of train-test splits on model performance in medical and remote sensing applications, highlighting the importance of balanced data splits for robust model evaluation (Pawluszek-Filipiak & Borkowski, 2020; Vrigazova, 2021; Rácz, Bajusz, & Héberger, 2021).
• The abstract could be more concise, focusing on key findings and their implications rather than detailed background information.
• Figures and tables, while informative, should be better integrated into the text to emphasize critical findings.
Code Review
Pros:
• The Jupyter Notebook presents a clear data pipeline, making it easy to follow the steps from data loading to model evaluation.
• The comparison of multiple models (Logistic Regression, Random Forest, KNN, SVC) across different train-test split ratios provides a comprehensive analysis.
• Visualization of results helps in understanding the impact of different train-test split ratios on model accuracy.
Cons:
• The MATLAB scripts are modular and clearly defined but lack detailed documentation and error handling.
• The Jupyter Notebook assumes the availability of correctly formatted data without including data preprocessing steps, which is crucial for reproducibility.

Experimental design

Manuscript Review
Pros:
• The choice of the BraTS 2013 dataset, a reputable dataset for brain tumor research, ensures the study's reproducibility and relevance.
• The comparison of multiple machine learning models provides a well-rounded perspective on the effect of train-test split ratios.
• The use of Principal Component Analysis (PCA) for feature reduction is a sound choice to handle high-dimensional data.
Cons:
• The rationale for selecting specific train-test split ratios (60:40 to 95:05) needs further justification. Including a discussion on how these ratios compare with other common splits used in similar studies, such as 70:30 or 80:20, would be beneficial (Vrigazova, 2021; Tan et al., 2021).
• The manuscript lacks details on the potential impact of data preprocessing steps on the results. More information on the implementation of the GLCM method and its effects on the dataset would add depth to the experimental design.
• The study could be improved by including cross-validation techniques to ensure the robustness of the results (Rácz, Bajusz, & Héberger, 2021).
Code Review
Pros:
• The Jupyter Notebook implements a straightforward and effective experimental design, including PCA for feature reduction and the comparison of different models.
• The MATLAB scripts for GLCM feature extraction are well-structured, providing a clear methodology for extracting relevant features from medical images.
Cons:
• The Jupyter Notebook could benefit from cross-validation to provide more reliable performance estimates.
• The MATLAB scripts lack error handling and feature scaling, which are important for ensuring robust and consistent performance.

Validity of the findings

Manuscript Review
Pros:
• The finding that higher training set percentages generally improve model performance aligns with established knowledge in machine learning (Pawluszek-Filipiak & Borkowski, 2020).
• The use of rank correlation to statistically analyze model accuracy across various train-test splits is robust and well-executed.
• Results are clearly presented in both tabular and graphical formats, making performance comparisons straightforward.
Cons:
• The manuscript could benefit from a more detailed discussion on the limitations of the study, such as the potential for overfitting with very high training set ratios (Pawluszek-Filipiak & Borkowski, 2020).
• Contextualizing the findings within the broader literature would help validate the conclusions. Comparing results with similar studies can provide additional support for the research outcomes (Tan et al., 2021; Vrigazova, 2021).
• The practical implications of the findings for medical image processing, including best practices for model development and deployment, should be elaborated upon.
Code Review
Pros:
• The results obtained from the Jupyter Notebook provide clear insights into how different train-test split ratios affect model performance.
• The MATLAB scripts effectively extract GLCM features, which are crucial for the subsequent analysis in the Jupyter Notebook.
Cons:
• The findings in the Jupyter Notebook could be validated further using cross-validation techniques to ensure robustness.
• More detailed documentation and explanation of the data preprocessing steps in both the Jupyter Notebook and MATLAB scripts would strengthen the validity of the findings.

Additional comments

Pros:
• The research addresses a relevant and underexplored issue in machine learning for medical image processing, offering valuable insights into train-test split ratios.
• The statistical analysis using Spearman's Rank Correlation Coefficient is a strong point, providing a detailed understanding of model performance relationships.
• Conclusions are well-supported by experimental results and provide clear directions for future research.
Cons:
• The manuscript would benefit from a more critical discussion of results, considering potential confounding factors and the generalizability of findings to other datasets and applications (Tan et al., 2021; Rácz, Bajusz, & Héberger, 2021).
• Language could be tightened to reduce redundancy, particularly in the introduction and discussion sections.
• Including more recent references and studies in the literature review would provide a more current perspective on the topic.

·

Basic reporting

- Line 75 onwards to next few lines, breakdown these 3 definitions into 3 bullets for easy read/refers

- In the Motivation section, I see some parts feel like repeated, especially the part which talks about Finding the right balance on train/test ration.

- 177 - extra line

- 192- space after fullstops

- 206 - blank line

- 259 - Mention what is BraTS 2013 dataset in earlier sections

- Mention ‘train-test splits’ consistency, somewhere it is training-test splits

- Mention number in 2 digit floating points e.g. 96.8 as 96.80 for easy readability

- Figure 3 - please explain x and y

Experimental design

-

Validity of the findings

-

Additional comments

Overall, very well articulated, with clear and concise English. I've shared some feedback to improve the readability of the language.

---

## Round 0.2 · accepted · Accept

Congratulations. The reviewer accepted your paper

·

Basic reporting

no comment

Experimental design

no comment

Validity of the findings

no comment

Additional comments

I'm happy with the new version, and do not have additional comments to share.